# Examining the extraction of parafoveal semantic information in Tibetan

**Meng Shen[1]☯, Zibei Niu[1]☯, Lei Gao[1], Tianzhi Li[1], Danhui Wang[1], Shan Li[1], Man Zeng[1], Xuejun Bai[2], Xiaolei Gao[1]***

1 Plateau Brain Science Research Center, Tibet University, Lhasa, China, 2 Key Research Base of Humanities and Social Sciences, Institute of Psychology and Behavior, Tianjin Normal University, Tianjin, China

☯ These authors contributed equally to this work.
* gaoxiaolei2010@163.com

**Data Availability Statement:** Data materials are available at: https://osf.io/dejv5/.

**Funding:** This work was supported by the National Natural Science Foundation of China (31860280, 32260204); the Cultivation Fund Project of Tibet

## Abstract

This study conducted two experiments to investigate the extraction of semantic preview information from the parafovea in Tibetan reading. In Experiment 1, a single-factor (preview type: identical vs. semantically related vs. unrelated) within-subject experimental design was used to investigate whether there is a parafoveal semantic preview effect (SPE) in Tibetan reading. Experiment 2 used a 2 (contextual constraint: high vs. low) × 3 (preview type: identical vs. semantically related vs. unrelated) within-subject experimental design to investigate the influence of contextual constraint on the parafoveal semantic preview effect in Tibetan reading. Supporting the E-Z reader model, the experimental results showed that in Tibetan reading, readers could not obtain semantic preview information from the parafovea, and contextual constraint did not influence this process. However, comparing high- and low-constrained contexts, the latter might be more conducive to extracting semantic preview information from the parafovea.

## Introduction

The human retina is divided into three regions: the foveal, parafoveal, and peripheral regions. The foveal (parafoveal) region is 1˚~ 2˚ (approximately 10˚) away from the reader's fixation point, and the peripheral region includes all areas except the foveal and parafoveal regions [1, 2].

In the process of reading, readers mainly obtain information from the fovea. However, the parafovea also plays an important role in information extraction [3–5]. The types of information that readers can obtain from the parafovea are the focus of current research [6, 7]. Investigating this topic not only allows a more detailed understanding of the essence of reading but also helps address the contradiction between the E-Z reader and SWIFT models regarding the allocation of attention resources in reading, that is, whether the distribution is sequential or parallel [8–14].

It is found that in Chinese reading, readers can obtain low-level information, such as morphology and phonology information [9, 10, 12, 15, 16], and high-level information such as semantic information from the parafovea [12, 17–21]. Wang et al. set up two preview conditions comprising semantic coherence and semantic violation using the boundary paradigm

University (ZDTSJH21-04); the Scientific Development funds for Local Region from the Chinese Government in 2022 (XZ202201YD0018C). The funders had no role in study design, data collection and analysis, decision to publish, or preparation of the manuscript.

**Competing interests:** The authors have declared that no competing interests exist.

[22]. The results showed that readers could obtain semantic preview information in Chinese reading. Yan et al. conducted an experiment using single-component characters in Chinese, further proving that readers could obtain semantic preview information in Chinese reading [12]. Furthermore, Yan and Kliegl used compound characters as experimental materials, and the results provided evidence that readers could extract semantic information from the parafovea [18].

In English reading, readers can extract the initial-letter information of words [23, 24], orthographic information [25, 26], abstract letter codes, and phonology information from the parafovea [27–30]. However, a disagreement remains on whether semantic preview information can be extracted [31–37]. Rayner et al. used the boundary paradigm to manipulate four preview conditions—identical (winter–winter), semantically related (winter–summer), completely unrelated (winter–length), and visually similar (winter–winken)—and investigated the extraction of semantic information from the parafovea in English reading [38]. The results showed no significant difference in terms of fixation duration in the semantically related versus completely unrelated conditions, indicating that semantic preview information could not be extracted from the parafovea. Rayner et al. replicated the aforementioned experiment, and the results also showed that semantic preview information could not be extracted from the parafovea in English reading [39]. However, some studies have found that in English reading, semantic preview information could be extracted from the parafovea. For example, Schotter adopted the boundary paradigm and set three preview conditions: identical with the target, synonyms, and semantically unrelated [40]. The results showed that the fixation duration under the semantically unrelated condition is significantly longer than that under the synonym condition, indicating that readers could extract semantic preview information from the parafovea under synonym conditions. Schotter and Jia set four preview conditions in their experiment: identical, antonyms or synonyms, reasonably semantically unrelated, and reasonably semantically related [36]. The results showed that the fixation duration in the reasonably semantically unrelated condition was significantly longer than that in the antonym or synonym condition, suggesting that readers could also extract semantic preview information from the parafovea in antonym conditions. Hohenstein et al. argued that, as compared to English, shallower orthographic-depth in German leads to faster phonological decoding, which in turn facilitates access to semantics [41]. This makes it possible for German readers to efficiently extract semantic knowledge from parafoveal words. However, their experiments used a combination of fast-priming and boundary-paradigm in which the parafoveal preview was presented only for a limited amount of time but not during the whole fixation on pre-target word. Subsequently, some researchers improved this. They obtained the evidence of semantic preview effect in German reading by using the standard boundary-paradigm [32].

The Tibetan language belongs to the alphabetic writing system, as well as the Tibeto–Burman language family of the Sino–Tibetan language family. It has alphabetic writing with thirty consonants, four vowel symbols, and five reverse letters as its basic character or word unit. Tibetan is written horizontally from left to right. Its written structure takes a consonant letter as the core, which is called the 'base word'. A 'base word' can form a word by itself. In some cases, multiple consonants are written above and below a 'base word' to form a word. Vowel symbols are added to the top, bottom, and centre of the consonant letter.

In addition to the unique word formation rules, the formation rules of Tibetan characters also have some distinct characteristics. The following are some examples:

1. A word is composed of a monosyllable, and in some cases, an integral word is composed of a meaningful monosyllable and an additional component. In addition, nouns and adjectives in Tibetan can be used as additional components, and most of them are adjectives;

2. Compound words are composed of monosyllabic morphemes;

3. New words are produced by adding a component behind monosyllabic words;

4. Polysyllabic words are composed of multiple meaningful monosyllabic words. Tibetan also uses the separators '᠂' and '|' as inter-word and inter-sentence markers. Compared with Chinese and English, Tibetan shows characteristics of both languages in terms of the language type, writing structure, character or word marking, and transparency. That is, Tibetan has characteristics of logographic and alphabetic script, which are not possessed by any other language [42, 43]. As shown in Table 1 [42, 43].

Given that Tibetan is such a unique language, as discussed above, what types of information could readers obtain from the parafovea in Tibetan reading? Research has confirmed that in Tibetan reading, readers could extract morphology and phonology information from the parafovea [43], but it is unclear whether they could extract semantic preview information.

Regarding the extraction of semantic information from the parafovea, researchers have found that contextual constraint could influence the semantic preview effect (SPE). Li et al. found that a low-constrained context is conducive to the extraction of semantic preview information in Chinese reading [17]. Meanwhile, Schotter et al. investigated the role of contextual constraint on the parafoveal SPE in English reading [44]. The results showed that a high-constrained context is conducive to the extraction of semantic information from the parafovea. Tibetan has the characteristics of both Chinese and English languages. As such, what influence does contextual constraint have on the extraction of semantic preview information in Tibetan reading?

In summary, to investigate the extraction of semantic preview information from the parafovea in Tibetan reading, this study conducted two experiments using an eye tracker, in which Tibet University undergraduates participated. Experiment 1 used the boundary paradigm and set up three types of preview words (identical, semantically related, and unrelated) to investigate the parafoveal SPE in Tibetan reading, if any. The boundary paradigm is applied to set an invisible boundary in front of the target word in a sentence. Specifically, before the reader's eyes crossed the boundary the target word was replaced by the preview word. After the reader's eyes crossed the boundary, the real target word appeared. With this paradigm, we can investigate the types and range of information obtained by a reader and the parafoveal processing of information [9, 45, 46]. The experimental hypothesis is that readers obtain semantic preview information, proposed as follows:

H1: The fixation duration in a semantically related condition is significantly shorter than that in an unrelated condition.

Experiment 2 also used the boundary paradigm to explore further the influence of contextual constraint on the parafoveal SPE in Tibetan reading. The experimental hypothesis is that a

**Table 1. Comparison of Tibetan with English and Chinese languages [42, 43].**

| Languages | Language type | | Structure | | Inter word mark | | Transparency | |
|---|---|---|---|---|---|---|---|---|
| | Alphabetic script | Logographic script | Linear structure | Stereoscopic quality | Character separation | Space | Transparent pronunciation | Opaque pronunciation |
| Tibetan | Y | | Y | Y | Y | | Y | |
| Chinese | | Y | | Y | | | | Y |
| English | Y | | Y | | | Y | Y | |

Y means yes, and N means no.

high-constrained context can promote the extraction of readers' semantic preview information, proposed as follows:

H2: In a high-constrained context, the fixation duration in a semantically related condition is significantly shorter than that in an unrelated condition.

If in a low-constrained context the fixation duration in a semantically related condition is significantly shorter than that in an unrelated condition, this would mean that a low-constrained context could promote the extraction of readers' semantic preview information.

## Experiment 1: Parafoveal semantic preview effect in Tibetan reading

Using the boundary paradigm, this study examined whether readers could extract semantic preview information from the parafovea in the process of Tibetan reading, that is, whether there is a parafoveal SPE in Tibetan reading.

### Methods

**Participants.** A total of 66 (33 males and 33 females) undergraduates of Tibet University, aged 20.53 years on average ($SD$ = 1.29), participated in this study. All participants were right-handed, with normal or corrected-to-normal vision, and no visual problems such as astigmatism or strabismus. They were Tibetan, and their native language was Tibetan. All participants signed informed consent before the experiment was conducted.

**Experimental design.** A single-factor (preview type: identical, semantically related, and unrelated) within-subject experimental design was adopted. The types of preview words were the independent variables and eye-movement measures were the dependent variables.

**Materials.** Tibetan is an alphabetical script. Following Schotter et al. [44], target words (42) and words related (42) and unrelated (42) in meaning to the target words were selected from The Modern Tibetan Frequency Dictionary, for a total of 126 words. The target, semantically related, and unrelated words corresponded to three preview conditions: identical, semantically related, and unrelated. No significant difference was found in the number of characters among the three preview words: $F$ (2, 125) = 0.40, $p$ > 0.05, and there was no significant difference in word frequency: $F$ (2, 125) = 0.62, $p$ > 0.05. See Table 2 for details. For each target word, a sentence frame was initially prepared, and then the target words in the sentence were replaced with semantically related words and unrelated words. Finally, 126 sentences were formed.

The following assessments of experimental materials were conducted.

1. Semantic relevance: Twenty participants who did not participate in the formal experiment rated the semantic relatedness between target and preview words on a 5-point scale (1 = completely irrelevant and 5 = very relevant). The relevance evaluation result for the target and semantically related words was $M$ = 4.06 ($SD$ = 1.22). The relevance evaluation

**Table 2. Means and standard deviations for word frequency and word length of three preview words.**

| Preview words | Identical | Semantically related | Unrelated |
|---|---|---|---|
| Word frequency (1/10000) | 4.91(8.77) | 6.82 (11.43) | 4.54 (9.61) |
| Word length (the Number of Characters) | 2.19 (0.92) | 2.33 (0.72) | 2.21 (0.68) |

Standard deviations are provided in parentheses.

result for the target and unrelated words was $M = 1.85$ ($SD = 1.16$). There was a significant difference between the two evaluation results ($t = 38.18$, $p < 0.001$).

2. Sentence naturalness: Twenty participants who did not participate in the formal experiment rated the sentence naturalness on a 5-point scale (1 = entirely unnatural and 5 = entirely natural). The evaluation result was $M = 4.28$ ($SD = 0.60$).

3. Sentence difficulty: Twenty participants who did not participate in the formal experiment rated the sentence difficulty on a 5-point scale (1 = very easy and 5 = very difficult). The evaluation result was $M = 1.72$ ($SD = 0.54$). The assessment results above showed that the experimental materials were appropriate for the experimental requirements.

Three files were constructed. Counterpart sentences from each set of three were allocated to one of the three files according to their experimental condition. Experimental conditions were rotated across files according to a Latin square. Each file included 42 experimental sentences of which 14 sentences were from each of the three conditions. All participants saw all of the 42 experimental sentences in a file, and each participant saw each sentence only once. In addition to the experimental sentences, 24 filler sentences were added to each block; 22 interrogative sentences were set to ensure that the participants read carefully, and all the sentences were presented randomly. Each participant only read one block. An invisible boundary was set on the left of the target word using the boundary paradigm. Before the reader's eye crossed the boundary, the preview word was presented. After the reader's eye crossed the boundary, the target word was presented. An example is shown in Fig 1.

**Apparatus.** An Eye-link 1000 plus eye tracker, produced by SR Research Ltd. (Ottawa, ON, Canada), was used in this study, and the sampling rate was 1,000 Hz. The refresh rate of the 21-inch CRT monitor (SONY MuLtiscanG520 from the Sony Group Corporation, Tokyo, Japan) was 140 Hz, and the resolution is 1,024 * 768 pixels. Participants were seated approximately 65 cm from the monitor. The stimulus was presented in Microsoft Himalaya 32 font, and each Tibetan character corresponded to approximately 0.6˚ of visual angle. All the experimental materials were presented on the screen with black words on a white background, and only one sentence was presented at a time.

*Procedures.* After the participants entered the lab, they were adjusted to the most comfortable sitting position in front of the monitor, and their chin was placed on the chin rest of the eye tracker. Before starting the experiment, the instructions were presented to the participants via the monitor. After the participants read the instructions, the researcher explained the experimental content and requirements and informed the participants of the positions of the page turning button and the "yes" and "no" judgment button. To ensure that the eye tracker accurately recorded the participants' eye movement trajectory, they completed a three-point calibration and validation procedure until they attained an average error below 0.25 degrees [43]. Recalibration and revalidation were conducted when necessary (i.e., when error increased beyond 0.25 degrees). After a successful calibration, the experiment started.

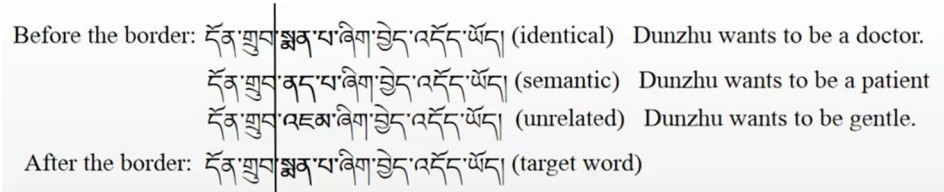

**Fig 1. Examples of experimental materials.**

**Dependent measures.** Referring to previous literature [43, 44], the first fixation duration (FFD), single fixation duration (SFD), gaze duration (GD), total fixation duration (TFD), and regression path reading time (RPRT) were selected as the dependent measures of the experiment.

FFD refers to the duration of the first fixation on the word, regardless of how many fixations are made. In the current research, FFD is one of the most commonly used dependent measures, which can effectively reflect the characteristics of the early stage of lexical access. SFD refers to the duration of a fixation on a word when it is the only fixation on that word in the first instance of reading. SFD also is an early reading time measure, which can effectively reflect the characteristics of the early stage of lexical access and is a good indicator of semantic activation in word recognition [47–49]. GD refers to the sum of all gaze durations before the fixation point leaves a region of interest for the first time. It also reflects the early stage of lexical access. TFD refers to the sum of all fixations on a word, including time spent re-reading the word after a regression back to it. TFD is sensitive to slower and longer cognitive processes. RPRT refers to the sum of the durations of all instances of fixation from the first fixation in an area of interest to that when the fixation point falls to the right of the areas of interest (excluding this fixation point). RPRT can not only reflect the process of lexical access, but also reflect the process of sentence integration in the later stage [48].

## Results

The linear mixed model and lme4 package in the R environment [50] were used to analyse the data [51]. The eye movement measures were log transformed before the linear mixed model was run. In the model, variables were designated as fixed factors and the participants, as crossing random effects, and the random intercept and random slope of participants and items were considered simultaneously [52]. To maximize the generalizability of our analyses, we used the maximal random effect structure [52] for all measures. If the maximum random model did not converge, the model was trimmed starting with removal of the correlation and slope of the items. Then remove the correlation and slope of the subjects until the model converged.

The average accuracy rate for all participants was 92.08%, indicating that the subjects read and understood the sentences carefully. Referring to the existing literature [21, 34, 43, 53], the data were eliminated according to the following criteria: (1) any fixation duration shorter than 80 ms or longer than 1,200 ms, (2) the data less than 5 fixation points during reading, (3) the data with fixation exceeding 3 standard deviations (the average deletion rate was 3.40%), and (4) trials in which there was a blink when the eye first passed through the boundary or fixed at the target word and trials in which the display change were premature or delayed (20.79%). See Table 3 for the means and standard deviation of each indication of the target word. And the significant difference of each measure is shown in Fig 2.

**Table 3. Means and standard deviation of measures in various conditions.**

| Dependent measures | Identical | Semantically related | Unrelated |
|---|---|---|---|
| FFD | 253 (56) | 282 (66) | 284 (75) |
| SFD | 261 (62) | 284 (76) | 291 (85) |
| GD | 343 (10) | 358 (93) | 372 (99) |
| TFD | 514 (180) | 576 (176) | 585 (191) |
| RPRT | 354 (117) | 399 (103) | 406 (101) |

Standard deviations are provided in parentheses. All measures are in milliseconds.

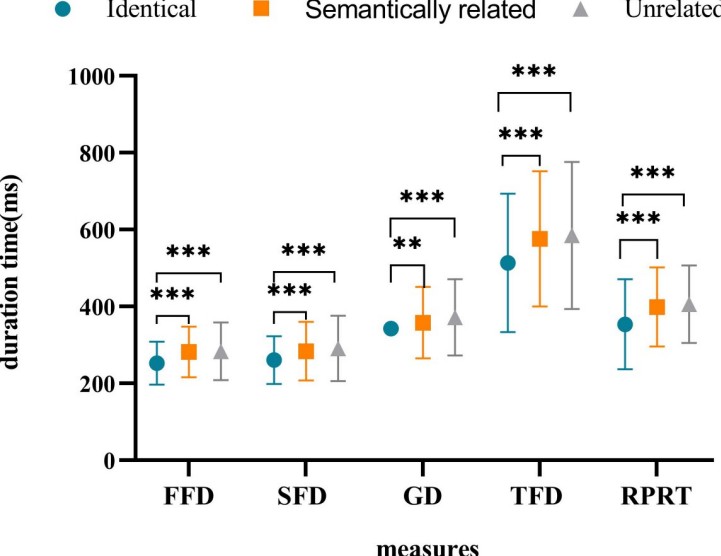

**Fig 2. The statistical result of experiment 1.**

Referring to relevant studies [21, 43], the target words were analysed as areas of interest. The results are summarized as follows:

1. The FFD in the identical condition was significantly shorter than that in the semantically related ($b = -0.109$, $SE = 0.025$, $t = -4.326$, $p < 0.001$) and unrelated ($b = -0.111$, $SE = 0.025$, $t = -4.468$, $p < 0.001$) conditions. There was no significant difference between the fixation time under the semantically related and unrelated conditions ($b = -0.003$, $SE = 0.025$, $t = -0.108$, $p = 0.914$);

2. The SFD in the identical condition was significantly shorter than that in the semantically related ($b = -0.097$, $SE = 0.030$, $t = -3.219$, $p = 0.001$) and unrelated ($b = -0.120$, $SE = 0.030$, $t = -4.038$, $p < 0.001$) conditions. There was no significant difference in the SFD under the semantically related versus unrelated conditions ($b = -0.023$, $SE = 0.030$, $t = -0.757$, $p = 0.449$);

3. The GD in the identical conditions was significantly shorter than that in the semantically related ($b = -0.082$, $SE = 0.030$, $t = -2.697$, $p = 0.007$) and unrelated ($b = -0.113$, $SE = 0.030$, $t = -3.770$, $p < 0.001$) conditions. There was no significant difference in the fixation time under the semantically related versus unrelated conditions ($b = -0.031$, $SE = 0.030$, $t = -1.037$, $p = 0.300$);

4. The TFD in the identical condition was significantly shorter than that in the semantically related ($b = -0.167$, $SE = 0.034$, $t = -4.962$, $p < 0.001$) and unrelated ($b = -0.163$, $SE = 0.033$, $t = -4.886$, $p < 0.001$) conditions. There was no significant difference in the fixation time under the semantically related versus unrelated conditions ($b = -0.004$, $SE = 0.034$, $t = 0.121$, $p = 0.903$); and

5. The RPRT in the identical condition was significantly shorter than that in the semantically related ($b = -0.135$, $SE = 0.032$, $t = -4.244$, $p < 0.001$) and unrelated ($b = -0.162$, $SE = 0.032$, $t = -5.132$, $p < 0.001$) conditions. There was no significant difference in the fixation time under the semantically related versus unrelated conditions ($b = -0.027$, $SE = 0.032$, $t = -0.847$, $p = 0.397$).

The results above indicate that in all measures, the fixation duration in the identical condition was significantly shorter than that in the semantically related and unrelated conditions, while there was no significant difference between the semantically related and unrelated conditions. These results in Tibetan reading suggest that readers could not obtain the semantic preview information from the parafovea, meaning that there was no SPE.

In view of the insignificant difference of all measures under the conditions of semantically related and unrelated word, the rstanarm package in R language [54] was used to conduct Bayesian analysis of linear mixed model for all measures. The prior distribution on the intercept was Normal (0, 15), and the prior distribution on the slopes was Normal (0, 1). Sampling from the posterior distribution was done with 5 Markov Chain Monte Carlo chains with 10,000 iterations each. The first 1,000 iterations were discarded as burn-in. Bayes factors were calculated using the Savage–Dickey density ratio method. Bayes factors greater than 1 favor the null hypothesis, while Bayes factors smaller than 1 favor the alternative hypothesis. The results showed that the BF of FFD, SFD, GZ, TFD, and RPRT was greater than 10 in the comparison of semantically related and semantically unrelated conditions (FFD:$BF = 32.24$, SFD: $BF = 38.27$, GZ:$BF = 26.42$, TFD:$BF = 27.06$, RPRT:$BF = 31.26$). There was strong evidence to support that there was no significant difference between the two conditions. A sensitivity analysis using a range of realistic priors indicated that the choice of prior did not influence the conclusions from this analysis.

## Experiment 2: The influence of context restriction on the parafoveal semantic preview effect in Tibetan reading

Using the boundary paradigm, this study investigated the influence of contextual constraint on the parafoveal SPE in Tibetan reading.

### Methods

**Participants.**　A total of 66 Tibet University undergraduates (32 males and 34 females), aged 20.70 years on average ($SD = 1.23$), participated in this experiment. All participants had normal or corrected-to-normal vision and did not have visual problems such as astigmatism or strabismus. They were all Tibetan, and their native language was Tibetan. All participants signed informed consent before the experiment was conducted.

**Experimental design.**　A 2 (contextual constraint: high vs. low) × 3 (preview type: identical vs. semantically related vs. unrelated) within-subject experimental design was adopted. Contextual constraint and preview type were the independent variables, and eye-movement measures were the dependent variable.

**Materials.**　The selection criteria were the same as in experiment 1. A total of 42 target words were initially selected, and then words related (42) and unrelated (42) to the meaning of the target words were selected individually, for a total of 126 words. The target, semantically related, and unrelated words corresponded to three preview conditions: identical, semantically related, and unrelated. There was no significant difference in the number of characters among the three preview words: $F (2, 125) = 1.85$, $p > 0.05$, and there was no significant difference in word frequency: $F (2, 125) = 0.62$, $p > 0.05$ (See Table 4 for details). Two sentence frames were prepared for one target word, one each for the high- and low-constrained contexts. The selection of the high- and low-constrained contexts was conducted as follows. First, the target words in the sentences were removed. Then, 18 Tibetan University undergraduates who did not participate in the formal experiment performed a cloze task. The results of the cloze predictability task showed that the close predictability under the high-constrained (low-constrained) context was 89% (5%). The difference was significant: $t = 51.88$, $p < 0.001$. Finally,

**Table 4. Means and standard deviations for word frequency and word length of the three preview words.**

| Preview word | identical | semantically related | unrelated |
|---|---|---|---|
| Word frequency (1/10000) | 4.85 (6.26) | 5.35 (9.07) | 7.17 (13.43) |
| Word length (the Number of Characters) | 2.38 (0.79) | 2.26 (0.73) | 2.10 (0.48) |

Standard deviations are provided in parentheses.

252 sentences were formed by replacing the target words in the sentence with semantically related and unrelated words.

The following assessments of experimental materials were conducted.

1. Semantic relevance: Twenty Tibetan university undergraduates who did not participate in the formal experiment rated the semantic relevance of the target, semantically related, and unrelated words on a 5-point scale (1 = completely unrelated and 5 = very related). The relevance evaluation results for the target and semantically related words were $M = 4.25$ ($SD = 1.09$), and that for the target and semantically unrelated words was $M = 1.98$ ($SD = 1.21$). There was significant difference between the two evaluation results ($t = 40.41$, $p < 0.001$).

2. Sentence naturalness: Twenty participants who did not participate in the formal experiment rated the sentence naturalness on a 5-point scale (1 = entirely unnatural and 5 = entirely natural), resulting in $M = 4.37$ ($SD = 0.67$).

3. Difficulty: Twenty Tibetan university undergraduates who did not participate in the formal experiment rated the sentence difficulty on a 5-point scale (1 = very easy and 5 = very difficult), resulting in $M = 1.68$ ($SD = 0.59$). The assessment results above showed that the experimental materials were appropriate for the experimental requirements.

The 252 sentences were divided into six blocks. There were 42 experimental sentences in each block. There were seven sentences each for the identical, semantically related, and unrelated conditions in the high- and low-constrained contexts. Each sentence context appeared only once in one block. In addition to the experimental sentences, 24 filler sentences were added to each block, and 22 question sentences were set to ensure that the participants read carefully. All sentences were presented randomly. Each participant only read one block.

**Apparatus, procedures, & dependent measures.** The experimental instruments, procedures, & dependent measures are the same as in Experiment 1.

## Results

The method of data analysis was the same as in Experiment 1. The average accuracy rate of all participants was 87.68%, indicating that the participants read and understood the sentences carefully. The standard of data deletion was identical to that in Experiment 1. Notably, the average deletion rate of the data whose fixation exceeded 3 standard deviations is 3.38%. Approximately 18.92% of the sentences—that blinked when the eye first passed the boundary or was fixed at the target word and for which the display changes were premature or delayed— were deleted. Table 5 presents the means and standard deviation of the eye movement measures on target words under different contexts. And the interaction between preview conditions and contextual constraints under each measure is shown in Figs 3–7.

Referring to relevant research [21, 43], the target word was analysed as areas of interest. The results are summarized as follow:

1. There was no significant difference in the FFD under the high- versus low-constrained contexts ($b = -0.037$, $SE = 0.034$, $t = -1.069$, $p = 0.286$). The FFD in the identical condition was

**Table 5. Means and standard deviation of eye movement measures on target words under different contexts.**

| Dependent variables | Context constraint | Preview | | |
|---|---|---|---|---|
| | | identical | semantically related | unrelated |
| FFD | high | 265 (91) | 278 (98) | 279 (93) |
| | low | 238 (84) | 298 (100) | 317 (111) |
| SFD | high | 278 (107) | 276 (123) | 296 (108) |
| | low | 280 (117) | 320 (106) | 358 (101) |
| GD | high | 336 (159) | 357 (165) | 363 (159) |
| | low | 357 (147) | 373 (172) | 373 (161) |
| TFD | high | 468 (225) | 534 (180) | 492 (190) |
| | low | 465 (222) | 589 (265) | 553 (214) |
| RPRT | high | 332 (123) | 359 (96) | 365 (92) |
| | low | 338 (95) | 372 (112) | 365 (103) |

Standard deviations are provided in parentheses. All measures are in milliseconds.

significantly shorter than that in the semantically related ($b = -0.140$, $SE = 0.042$, $t = -3.338$, $p < 0.001$) and unrelated ($b = -0.152$, $SE = 0.042$, $t = -3.631$, $p < 0.001$) conditions. There was no significant difference in the FFD under the semantically related versus unrelated conditions ($b = -0.013$, $SE = 0.042$, $t = -0.298$, $p = 0.286$). There was a significant interaction of preview conditions (identical and unrelated) and contextual constraint ($b = -0.169$, $SE = 0.084$, $t = -2.012$, $p = 0.045$). Further analysis showed that only in the low-constrained context condition was the FFD in the identical condition significantly shorter than that in the semantically unrelated condition ($b = -0.168$, $SE = 0.072$, $t = -2.346$, $p = 0.019$). The interaction of contextual constraint and preview conditions (semantically related and unrelated) was not significant ($b = -0.008$, $SE = 0.085$, $t = -0.100$, $p = 0.921$).

2. The SFD in the high-constrained context condition was significantly shorter than that in the low-constrained context condition ($b = -0.163$, $SE = 0.055$, $t = -2.971$, $p = 003$). The

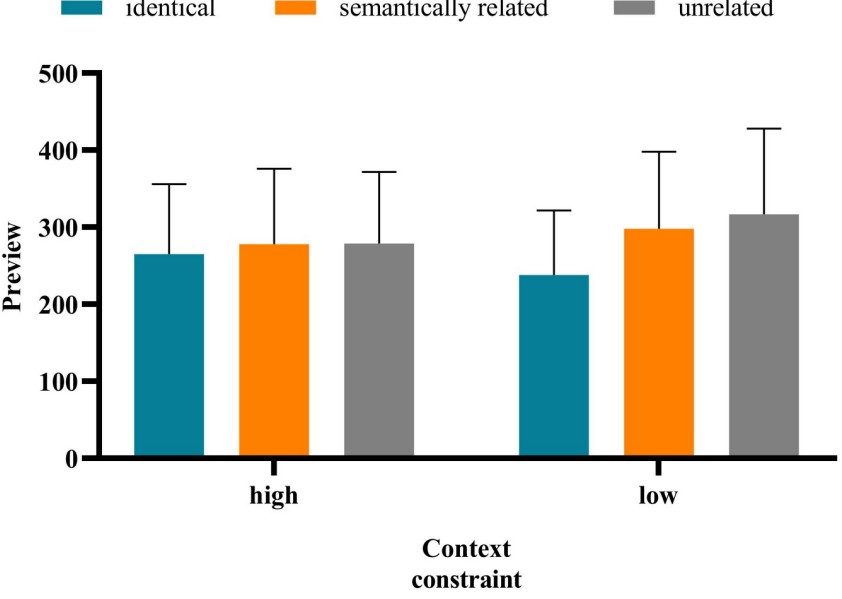

**Fig 3.**

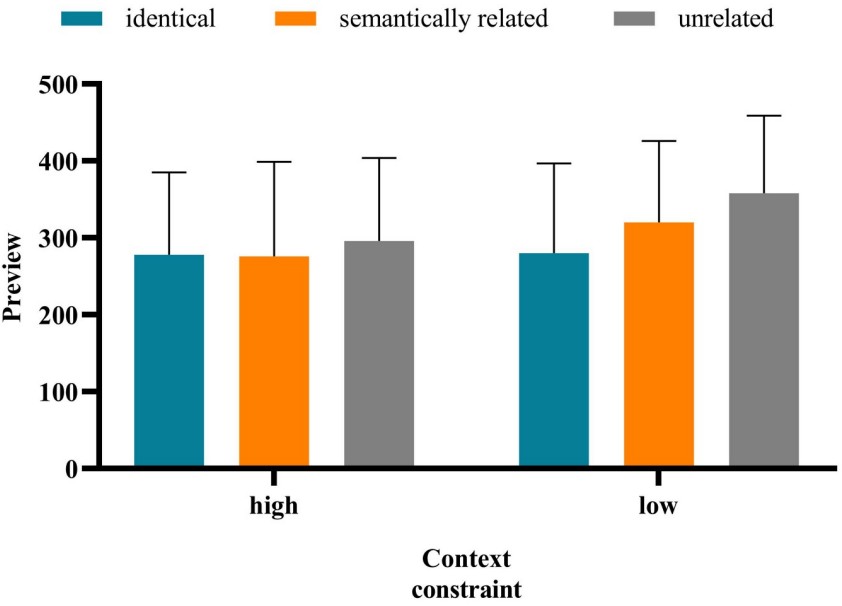

**Fig 4.**

SFD in the identical condition was significantly shorter than that in the unrelated condition ($b = -0.144$, $SE = 0.066$, $t = -2.197$, $p = 0.030$). Moreover, there was no significant difference in the SFD under the semantically related versus unrelated conditions ($b = -0.042$, $SE = 0.067$, $t = -0.635$, $p = 0.527$). The interaction between contextual constraint and preview conditions was not significant.

3. There was no significant difference in GD under the high- versus low-constrained context conditions ($b = -0.048$, $SE = 0.043$, $t = -1.132$, $p = 0.265$). There was no significant

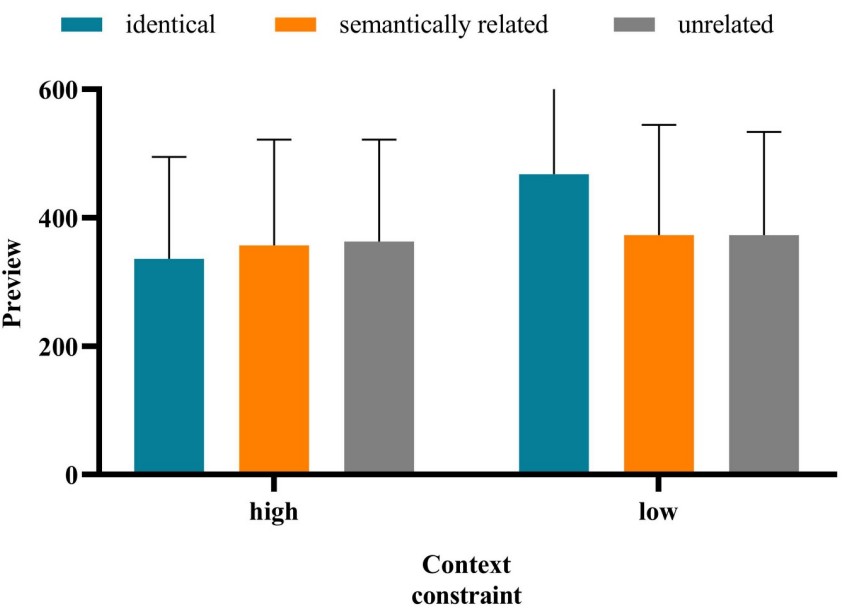

**Fig 5.**

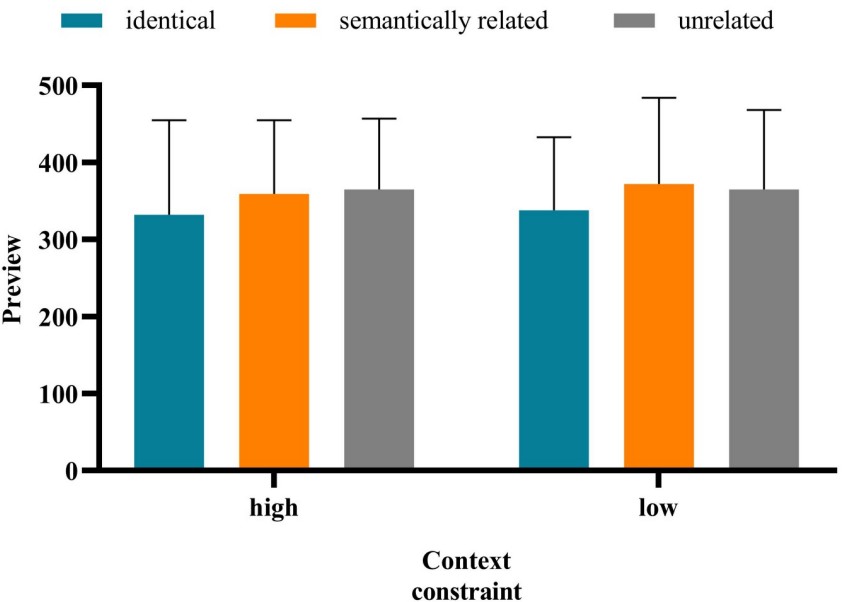

**Fig 6.**

difference in GD under the semantically related ($b$ = -0.070, $SE$ = 0.052, $t$ = -1.360, $p$ = 0.183) versus unrelated conditions ($b$ = −0.066, $SE$ = 0.052, $t$ = −1.286, $p$ = 0.206). There was no significant difference in GD under the semantically related versus unrelated conditions ($b$ = 0.004, $SE$ = 0.052, $t$ = 0.071, $p$ = 0.943). The interaction between contextual constraint and preview conditions was not significant.

4. There was no significant difference in the TFD under the high- versus low-constrained context conditions ($b$ = -0.043, $SE$ = 0.038, $t$ = -1.134, $p$ = 0.264). The TFD in the identical

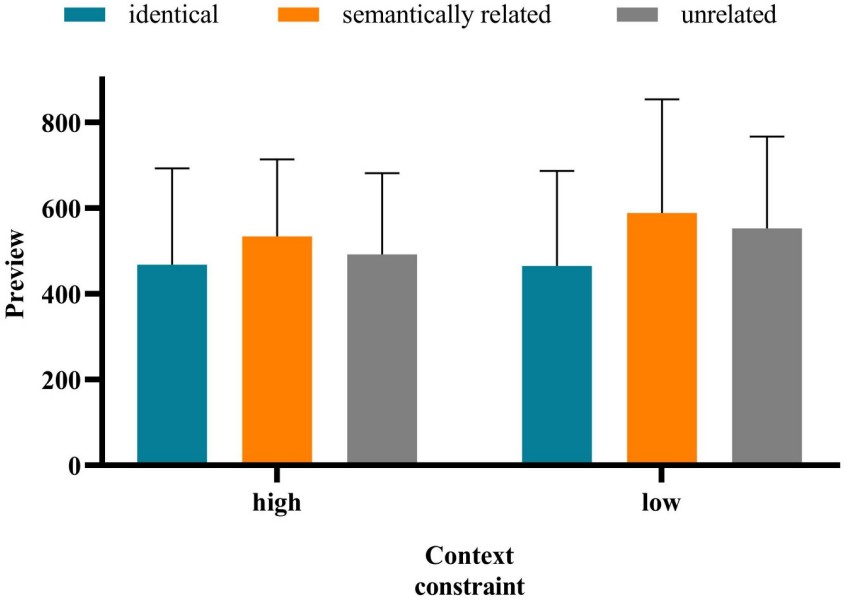

**Fig 7.**

condition was significantly shorter than that in the semantically related ($b = −0.197$, $SE = 0.046$, $t = −4.271$, $p < 0.001$) and unrelated ($b = −0.147$, $SE = 0.046$, $t = −3.185$, $p = 0.003$) conditions. There was no significant difference in the TFD under the semantically related versus unrelated ($b = 0.050$, $SE = 0.046$, $t = 1.087$, $p = 0.284$) conditions. The interaction between contextual constraint and preview conditions was not significant.

5. There was no significant difference in the RPRT under the high- versus low-constrained contexts ($b = −0.006$, $SE = 0.024$, $t = −0.266$, $p = 0.791$). The RPRT in the identical condition was significantly shorter than that in the semantically related ($b = −0.086$, $SE = 0.029$, $t = −2.94$, $p = 0.003$) and unrelated ($b = −0.077$, $SE = 0.029$, $t = −2.615$, $p = 0.009$) conditions. There was no significant difference in the RPRT under the semantically related versus unrelated conditions ($b = 0.009$, $SE = 0.029$, $t = 0.319$, $p = 0.750$). The interaction between contextual constraint and preview conditions was not significant.

The results above indicate that the SFD in the high-constrained context condition was significantly shorter than that in the low-constrained context condition. The SFD is an early reading time measure, which can effectively reflect the characteristics of the early stage of lexical access and is a good indicator of semantic activation in word recognition [47–49]. Therefore, to a certain extent, the aforementioned results showed that contextual constraint has an influence on the process of Tibetan reading, and occurs in the early stage of vocabulary processing. In all measures, there was no significant difference in the fixation duration under the semantically related versus unrelated conditions, indicating that in Tibetan reading, readers do not obtain semantic preview information from the parafovea. This finding is consistent with that of Experiment 1. For the FFD, the interaction between contextual constraint and preview conditions (identical and unrelated) was significant. Only in the low-constrained context condition was the FFD in the identical condition significantly shorter than that in the unrelated condition. However, in the low-constrained context condition, there was no significant difference in the FFD under the semantically related versus unrelated conditions. These results indicate that in Tibetan reading, contextual constraint has no influence on the extraction of semantic information from the parafovea.

In order to provide further statistical support for the null interaction of contextual constraint and preview conditions, we undertook Bayes factor analyses for linear mixed models [55] in relation to FFD, SFD, GZ, TFD, and RPRT. Bayes factors both for the full model (i.e., BF-Full, the model containing the main effects of contextual constraint and preview conditions, and their interaction) and the model with only main effects (i.e., BF-base) were calculated. By comparing the two models (BF = BF-Full/BF- base), we were able to evaluate the nonsignificant interaction between contextual constraint and preview conditions. BF values smaller than 1 favor the null hypothesis, whereas BF values greater than 1 favor the alternative hypothesis. For each of the reading time measures we used the default scale prior ($r = 0.5$) and 100,000 Monte Carlo iterations of the Bayes-Factor package. The results of Bayesian analysis favored the null hypothesis (FFD: $BF = 0.27$, SFD: $BF = 0.27$, GZ: $BF = 0.08$, TFD: $BF = 0.08$, RPRT: $BF = 0.03$). Also, a sensitivity analysis with different priors (i.e., 0.2, 0.3, 0.4, 0.5, 0.6, 0.7, and 0.8) provided consistent results (all $BF < 0.60$).

## Discussion

### Semantic preview effect in Tibetan reading

Tibetan University undergraduates participated in Experiment 1, which employed three types of preview words (identical, semantically related, and unrelated). This study investigated whether readers could extract semantic preview information from the parafovea in the process

of Tibetan reading. That is, whether there was a parafoveal SPE in Tibetan reading. It was hypothesised that the fixation duration in semantically related conditions is significantly shorter than that in unrelated conditions; in other words, readers obtain semantic preview information and therefore, there is an SPE in Tibetan reading.

The results showed that there was no significant difference in all measures under the semantically related versus unrelated conditions, indicating that readers could not obtain semantic preview information in the process of Tibetan reading. In other words, there was no SPE in Tibetan reading. This result was also confirmed in Experiment 2.

The possible reasons for the aforementioned results are as follows:

1. The mapping from orthography to semantics affects the extraction of semantic preview information [39]. Specifically, a direct mapping from orthography to semantics promotes the extraction of semantic information. By contrast, if there is no direct mapping, the possibility of extracting semantic preview information is smaller [4, 12]. As an alphabetic writing system, Tibetan has the rules of orthography–phonology correspondence [56], in which a direct mapping from orthography to phonology exists; however, there is often no direct mapping from orthography to semantics. Thus, in Tibetan reading, the parafoveal processing is sensitive to phonological but not sensitive to semantic information. This characteristic is not conducive to the extraction of semantic preview information;

2. Orthographic information affects the extraction of semantic preview information. Studies have found that in an alphabetic writing system, salient orthographic information uses less attention resources, reduces the cost of vocabulary processing, and allows greater semantic access [32, 39]. Tibetan is not only an alphabetic writing; it is also classified as a Sino–Tibetan language. Therefore, Tibetan is closely related to Chinese, especially in terms of writing structure. In addition to the linear structure, Tibetan has the characteristic of multiple consonants being added around a 'base word' and written up and down to form a word [43]. Therefore, Tibetan orthography is more complex than a traditional alphabetic writing system. The complexity may require more attention resources and increase the cost of vocabulary processing. Consequently, the extraction of semantic information in Tibetan reading is inhibited;

3. Obvious boundary information between words affects the extraction of semantic preview information. Many studies have found that it is more difficult for readers to read a text without spaces than a text with normal spaces (reading a text without spaces is about 40% to 70% slower than reading a text with normal spaces) [31, 57, 58]. Tibetan language has word separations, but in contrast, the spacing between words is less apparent to readers, lacking visual clues of obvious word boundaries. Therefore, when reading Tibetan, readers may use more attention resources for word segmentation, and the process of word recognition will be hindered, unable to obtain high-level semantic information from parafoveal words; and

4. The E-Z reader model holds that semantic access in word recognition is a sequential process. When reading, readers only process one word at a time. Word N is processed first, and then word N + 1 is processed. When fixed at word N, readers can only extract the low-level information of word N + 1 (such as phonology, orthographic information, etc.), but cannot extract high-level information (such as semantic information) [7, 59]. The results of this study support the E-Z reader model.

The results of this study are also consistent with those of previous studies [38, 39, 60]. For instance, Rayner et al. first used the boundary paradigm to investigate whether readers could

obtain semantic preview information in English from the parafovea [38]. The results showed that readers could not obtain semantic preview information. Rayner et al. repeated the afore-mentioned experiments in 2014 [39], using the same experimental materials as in Rayner et al. [38], and obtained the same results. Altarriba et al. chose English and Spanish bilinguals as participants and used the boundary paradigm to manipulate four preview conditions in one of their experiments [60]: translation, semantically related non-cognate, pseudo-cognate, and semantically unrelated words. Pseudo-cognate words referred to words that were similar to the orthography of the target word but had no semantic relationship in another language. The results showed that there was no significant difference in fixation time between pseudo-cognate and translations words, indicating that readers could not obtain semantic preview information in reading.

## The influence of contextual constraint on the parafoveal semantic preview effect in Tibetan reading

Tibetan University undergraduates also participated in Experiment 2, in which the contextual constraint and preview conditions were manipulated, and the contextual constraint was divided into high- and low-constrained contexts. The preview conditions were in line with Experiment 1. Experiment 2 was conducted to explore the influence of contextual constraint on the extraction of semantic preview information in Tibetan reading. The experimental hypothesis was that in a high-constrained context, the fixation duration of semantically related conditions is significantly shorter than that in unrelated conditions; in other words, a high-constrained context can promote the extraction of semantic preview information in reading. If in low-constrained context, fixation duration in the semantically related conditions is significantly shorter than that in the unrelated conditions, this would indicate that a low-constrained context could promote the extraction of a reader's semantic preview information.

The results of Experiment 2 showed that in Tibetan reading, the SFD in a high-constrained context was significantly shorter than that in a low-constrained context, indicating that a high-constrained context is more conducive to Tibetan reading. Moreover, the activation of contextual constraint on vocabulary occurred in the early stage of lexical processing. It was found that the overall context of sentences would activate or constrain the features of the target word [61]. The more the vocabulary features that were activated, the smaller the scope of lexical activation and the easier was the lexical processing. By contrast, the fewer the lexical features that were activated, the greater the scope of lexical activation and the more difficult was the lexical processing. Different constrained contexts have different degrees of constraints on vocabulary. Compared with a low-constrained context, a high-constrained context had greater limits on target words, activated a smaller scope, and had more features of the target words. As such, the target words were more easily integrated and understood, which promoted the processing of these target words [53]. These results are consistent with previous research results indicating that contextual information could promote lexical processing in different languages [25, 34, 44, 62–66].

The results of Experiment 2 also showed that the interaction between contextual constraint and preview conditions (semantically related and unrelated) was not significant. In both high- and low-constrained contexts, there was no significant difference in the fixation duration under the semantically related versus unrelated conditions, indicating that contextual constraint did not affect the extraction of semantic preview information. However, in terms of the two early reading time measures (FFD and SFD), the fixation time difference between semantically related and unrelated conditions in the low-constrained context condition ($FFD_{\text{unrelated–semantically related}}$ = 16 ms; $SFD_{\text{unrelated–semantically related}}$ = 38 ms) was greater than

that in the high-constrained context condition ($FFD$ unrelated–semantically related = 2 ms; $SFD$ unrelated–semantically related = 20 ms). It was speculated that a low-constrained context might be more conducive to the extraction of semantic preview information in Tibetan reading.

The possible reasons for the aforementioned results are as follows:

1. Previous studies suggested that sentence constraint leads readers to generate expectations about upcoming words, rather than make a specific, unitary prediction of the upcoming word [67–69]. A high-constrained context provides readers with sufficient information, which, to a certain extent, leads to some expectations for the target word. Once the readers gaze at the target word, it will be matched with the expected word. If they cannot be matched, there will be interference. Therefore, when semantically related words appear in the area of the target word, readers will find that the actual words (preview words) are inconsistent with the expected words, and they cannot be matched, which results in interference. Consequently, readers cannot extract semantic preview information. A low-constrained context provides less information to readers. Therefore, they will have no expectation for the target word and will focus on the lexical information provided. When semantically related preview words appear in the area of the target word, readers will not match the preview words with the target words, but will use lexical information to complete the integration of information. Therefore, a low-constrained context may promote readers to obtain semantic preview information, and

2. The preview time from the parafoveal will affect the semantic preview effect. In a short preview time, semantically related preview words will promote the processing of target words and show an SPE. Meanwhile, a long preview time may lead to inappropriate integration for semantics. Once the target word replaces the preview word, it will interfere with the processing of the target word and eliminate the SPE [17, 70]. A high-constrained context accelerates the extraction of semantic preview information and triggers the interference that can only be found in a long-term preview. The occurrence of interference for an SPE is relatively early. Consequently, there is no SPE. By contrast, a low-constrained context does not accelerate information extraction and does not show or delay the occurrence of hindrance to the SPE. Therefore, readers may extract semantic preview information in a low-constrained context. In summary, compared with a high-constrained context, a low-constrained context may be more conducive to extracting semantic preview information in Tibetan reading.

## Conclusion

This study explored the extraction of semantic preview information from the parafovea in Tibetan reading and the influence of contextual constraints on the extraction of semantic information. The experimental results showed that in Tibetan reading, readers could not extract semantic preview information from the parafovea, thus supporting the E-Z reader model. The results also revealed that contextual constraints do not influence the extraction of semantic preview information in Tibetan reading. However, compared with a high-constrained context, a low-constrained context might be more conducive to extracting the semantic preview information from the parafovea. The results of this study show that the semantic preview effect does not exist in eye-movement studies, but this does not rule out the possibility of its existence in neural effects. In the future, whether (and to what extent) Tibetan readers can extract high-level semantic preview information may require more direct brain activity measurement (such as ERPs). Our future research will continue to focus on this point.

## Acknowledgments

Many thanks to all the participants who for anonymity protection reasons are not mentioned here by name. Without their generosity in spending their time in the lab with no compensation and their willingness to share something so tremendously intimate to all of them, the present study would not have been possible.

## Author Contributions

**Conceptualization:** Meng Shen, Zibei Niu, Xiaolei Gao.

**Data curation:** Zibei Niu.

**Formal analysis:** Meng Shen, Danhui Wang, Shan Li, Man Zeng.

**Funding acquisition:** Xiaolei Gao.

**Investigation:** Meng Shen.

**Methodology:** Meng Shen, Zibei Niu.

**Project administration:** Lei Gao, Xiaolei Gao.

**Software:** Zibei Niu, Tianzhi Li.

**Supervision:** Lei Gao, Tianzhi Li, Xuejun Bai, Xiaolei Gao.

**Visualization:** Zibei Niu.

**Writing – original draft:** Meng Shen, Zibei Niu.

**Writing – review & editing:** Zibei Niu, Xiaolei Gao.

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
