## [Decision Letter · Decision Letter 0]

12 Dec 2022

PONE-D-22-28563Examining the extraction of parafoveal semantic information in TibetanPLOS ONE

Dear Dr. Gao,

I firstly want to apologize for getting you this verdict so late. It proved difficult to find reviewers. Therefore, in addition to one external reviewer, I've decided to carefully assess your manuscript myself, as I happen to have some expertise in this field.

     Both the Reviewer and I are of the opinion that this is an interesting and well-conducted study. I also find your paper to be well-written. Thus, I'm confident that we can proceed to accept your paper for publication, pending a few minor revisions. Please have a look at the suggestions for improvement noted by the Reviewer below. I agree with the Reviewer that, ultimately, the question of whether (and to what extent) higher-order information can be extracted from upcoming words may not be solved without involving more direct measures of brain activity (e.g. EEG). This has been discussed in recent years (see e.g. Snell & Grainger, 2019; Schotter & Payne, 2019, *TiCS*) and would be worth reflecting upon.        I'm also interested to know what you think about the fact that the Tibetan writing system doesn't comprise inter-word spacing (or at least, the boundaries between words seem visually less apparent to me). Perhaps you can reflect on whether this may lead to a different allocation of spatial attention when reading Tibetan as compared to a writing system that does comprise inter-word spacing. This might also explain the discrepancy between your study and the growing number of studies that do observe parafoveal semantic processing (or perhaps you can think of different ways to account for the discrepancy).

We look forward to receiving your revised manuscript.

Kind regards,

Joshua Snell, PhD

Academic Editor

PLOS ONE

Journal Requirements:

2. PLOS ONE does not copy edit accepted manuscripts (https://journals.plos.org/plosone/s/criteria-for-publication#loc-5). To that effect, please ensure that your submission is free of typos and grammatical errors.

"The funders had no role in study design, data collection and analysis, decision to publish, or preparation of the manuscript"

Reviewers' comments:

Reviewer's Responses to Questions

**Comments to the Author**

1. Is the manuscript technically sound, and do the data support the conclusions?

Reviewer #1: Yes

2. Has the statistical analysis been performed appropriately and rigorously? 

Reviewer #1: Yes

3. Have the authors made all data underlying the findings in their manuscript fully available?

Reviewer #1: Yes

4. Is the manuscript presented in an intelligible fashion and written in standard English?

Reviewer #1: Yes

5. Review Comments to the Author

Reviewer #1: The paper reports two eye-tracking experiments using the boundary paradigm to investigate parafoveal semantic preview effects in the reading of Tibetan. The unique writing system of Tibetan provides an opportunity to bridge the gap between orthographic and logographic writing systems. The authors find that the semantic relatedness of an upcoming word does not impact any reading time measures. They note, however, that the contextual constraints on the word may facilitate semantic preview in low-constrained contexts. They conclude that the absence of semantic relatedness effects provides support for the serial E-Z reader model of reading.

Overall, this is a well-conducted study of an interesting topic in the field of reading. The authors have taken great care in the design of their experiments and statistical analysis. I applaud their use of both frequentist and Bayesian statistics in their analysis. I provide some brief suggestions below:

• Given the perk of Bayesian statistics of being able to quantify evidence for a null hypothesis, and that the bulk of the reported effects in the paper are indeed null, I think leveraging this fact to support the lack of difference between semantic relatedness conditions would strengthen the analysis.

• I think it bears noting in either the literature review or the discussion that some effects which are not observable in behavior may still be observable in ERPs. The paper presents strong evidence that indeed the effect is absent in eye-movements, however this unfortunately does not preclude the possibility that a neural effect still exists.

• Consider including figures to illustrate effects (or lack thereof)

• In table 1, consider leaving blanks instead of Ns for easier reading

6. PLOS authors have the option to publish the peer review history of their article (what does this mean?). If published, this will include your full peer review and any attached files.

Reviewer #1: No

---

## [Author Response · Author response to Decision Letter 0]

27 Dec 2022

Dear Editor：

First of all, we would like to thank you very much for your comments. 

This comment is very valuable to us. You proposed to explain our results by the fact that there is no obvious space information in the Tibetan reading, which we have modified and marked in blue in the revised manuscript. The specific content is as follows: “Obvious boundary information between words affects the extraction of semantic preview information. Many studies have found that it is more difficult for readers to read a text without spaces than a text with normal spaces (reading a text without spaces is about 40% to 70% slower than reading a text with normal spaces) [31,58,59]. Tibetan language has word separations, but in contrast, the spacing between words is less apparent to readers, lacking visual clues of obvious word boundaries. Therefore, when reading Tibetan, readers may use more attention resources for word segmentation, and the process of word recognition will be hindered, unable to obtain high-level semantic information from parafoveal words; and”

Due to the modifications that we have made in the manuscript text, we have added corresponding references. Specially: “

58. Veldre A, Drieghe D, Andrews S. Spelling ability selectively predicts the magnitude of disruption in unspaced text reading. J Exp Psychol Hum Percept Perform. 2017;43. doi:10.1037/xhp0000425

59. Rayner K, Pollatsek A. Reading unspaced text is not easy: comments on the implications of Epelboim et al.’s (1994) study for models of eye movement control in reading. Vision research. 1996. doi:10.1016/0042-6989(95)00132-8”

Secondly, we adjusted the format of the manuscript and financial support according to the formatting-sample you provided.

Finally, we have made corresponding modifications to the comments made by the Reviewer, and attached a response below.

Response to Reviewers

Reviewer # 1: Review Comments to the Author

The paper reports two eye-tracking experiments using the boundary paradigm to investigate parafoveal semantic preview effects in the reading of Tibetan. The unique writing system of Tibetan provides an opportunity to bridge the gap between orthographic and logographic writing systems. The authors find that the semantic relatedness of an upcoming word does not impact any reading time measures. They note, however, that the contextual constraints on the word may facilitate semantic preview in low-constrained contexts. They conclude that the absence of semantic relatedness effects provides support for the serial E-Z reader model of reading.

Answer: First, I would like to express our sincere gratitude to the reviewer for their comments. These comments are all valuable and helpful for revising and improving our manuscript, as well as the important guiding significance to our research. We have studied comments and have made correction which we hope meet with approval. Revised portions are marker in blue in the revised manuscript. The summary of corrections and the responses to the reviewer’s listed below.

1.Comments：Given the perk of Bayesian statistics of being able to quantify evidence for a null hypothesis, and that the bulk of the reported effects in the paper are indeed null, I think leveraging this fact to support the lack of difference between semantic relatedness conditions would strengthen the analysis.

Answer: Thank you for your comment. This comment is very valuable to us. And we have made changes according to your comments. The specific content is as follows: “In view of the insignificant difference of all measures under the conditions of semantically related and unrelated word, the rstanarm package in R language [55] was used to conduct Bayesian analysis of linear mixed model for all measures. The prior distribution on the intercept was Normal (0, 15), and the prior distribution on the slopes was Normal (0, 1). Sampling from the posterior distribution was done with 5 Markov Chain Monte Carlo chains with 10,000 iterations each. The first 1,000 iterations were discarded as burn-in. Bayes factors were calculated using the Savage–Dickey density ratio method. Bayes factors greater than 1 favor the null hypothesis, while Bayes factors smaller than 1 favor the alternative hypothesis. The results showed that the BF of FFD, SFD, GZ, TFD, and RPRT was greater than 10 in the comparison of semantically related and semantically unrelated conditions (FFD:BF = 32.24，SFD:BF = 38.27，GZ:BF = 26.42，TFD:BF = 27.06，RPRT:BF = 31.26). There was strong evidence to support that there was no significant difference between the two conditions. A sensitivity analysis using a range of realistic priors indicated that the choice of prior did not influence the conclusions from this analysis.”

Due to the modifications that we have made in the manuscript text, we have added corresponding references. Specially: “

55. Vasishth S, Nicenboim B. Statistical Methods for Linguistic Research: Foundational Ideas – Part I. Lang Linguist Compass. 2016;10. doi:10.1111/lnc3.12201”

2.Comments：I think it bears noting in either the literature review or the discussion that some effects which are not observable in behavior may still be observable in ERPs. The paper presents strong evidence that indeed the effect is absent in eye-movements, however this unfortunately does not preclude the possibility that a neural effect still exists.

Answer: Thank you for your comment. This comment is very valuable to us. According to the suggestion, we have revised the corresponding contents in the revised manuscript. The specific content is as follows: “The results showed that the semantic preview effect does not exist in eye-movements, but this does not rule out the possibility of its existence in neural effect. In the future, the question of whether (and to what extent) higher-level information can be extracted from parafoveal words in Tibetan reading may require more direct brain activity measurement (such as ERPs), and our future research will continue to focus on this topic.”

3.Comments：Consider including figures to illustrate effects (or lack thereof).

Answer: Thank you for your comment. We apologize for not provide figures to illustrate effects (or lack thereof). And we have made changes according to your suggestions. The specific content is as follows: Fig 2. is for the statistical result of experiment 1, and Fig 3-7. is for the statistical result of experiment 2.

Fig 2. the statistical result of experiment 1

Fig 3. FFD Fig 4. SFD 

Fig 5. GD Fig 6. TFD

Fig 7. RPRT

4.Comments：In table 1, consider leaving blanks instead of Ns for easier reading.

Answer: Thank you for your comment. We have modified it based on your suggestions. The specific content is as follows: 

Table 1. Comparison of Tibetan with English and Chinese Languages.

Languages Language type Structure Inter word mark Transparency

 Alphabetic script Logographic script Linear structure Stereoscopic quality Character separation 

Space

 Transparent pronunciation Opaque pronunciation

Tibetan Y Y Y Y Y 

Chinese Y Y Y

English Y Y Y Y

---

## [Editor Report · Decision Letter 1]

27 Jan 2023

Examining the extraction of parafoveal semantic information in Tibetan

PONE-D-22-28563R1

Dear Dr. Gao,

We’re pleased to inform you that your manuscript has been judged scientifically suitable for publication and will be formally accepted for publication once it meets all outstanding technical requirements.

Kind regards,

Joshua Snell, PhD

Academic Editor

PLOS ONE
---

## [Editor Report · Acceptance letter]

8 Feb 2023

PONE-D-22-28563R1 

Examining the extraction of parafoveal semantic information in Tibetan 

Dear Dr. Gao:

I'm pleased to inform you that your manuscript has been deemed suitable for publication in PLOS ONE. Congratulations! Your manuscript is now with our production department. 

Kind regards, 

on behalf of

Dr. Joshua Snell 

Academic Editor

PLOS ONE